# Hypothalamic Median Eminence Thyrotropin-Releasing Hormone-Degrading Ectoenzyme Activity Is Dispensable for Basal Thyroid Axis Activity in Lean Rodents

**DOI:** 10.3390/cells14100725

**Published:** 2025-05-15

**Authors:** Adair Rodríguez-Rodríguez, Rosa María Uribe, Antonieta Cote-Vélez, Patricia Joseph-Bravo, Jean-Louis Charli

**Affiliations:** Departamento de Genética del Desarrollo y Fisiología Molecular, Instituto de Biotecnología, Universidad Nacional Autónoma de México (UNAM), Avenida Universidad 2001, Cuernavaca 62210, Mexico; adair.rodriguez@ibt.unam.mx (A.R.-R.); rosa.uribe@ibt.unam.mx (R.M.U.); antonieta.cote@ibt.unam.mx (A.C.-V.); patricia.joseph@ibt.unam.mx (P.J.-B.)

**Keywords:** *Trh*, *Trhde*, *Trhr*, TSH, thyroid hormone, hypothalamus, median eminence, tanycyte, portal capillaries, pituitary

## Abstract

The amplitude of the phasic output of thyrotropin-releasing hormone (TRH) into the hypothalamus–pituitary portal capillaries is likely controlled by the TRH-degrading ectoenzyme (TRH-DE) expressed on the surface of median eminence (ME) β2-tanycytes. To extend this hypothesis, we performed experiments on adult rodents reared in standard conditions. TRH-DE was close to the putative sites of TRH release in the male rat external layer of the ME. In global *Trhde* knockout mice, basal hypothalamus–pituitary–thyroid (HPT) axis parameters were not altered but we detected an increased vimentin (a tanycyte marker) positive coverage of the portal vessels. We then overexpressed TRH-DE or a dominant negative isoform by microinjection of adeno-associated virus 1 (AAV1) vectors into the third ventricle of adult male rats. Two weeks after microinjection, cold-stress-induced serum TSH concentration was decreased if ME TRH-DE activity had been enhanced. However, the long-term modification of TRH-DE activity in the ME had only a small impact on basal serum TSH concentration but increased *Trhr* expression in the anterior pituitary of animals transduced with AAV1-TRH-DE. Thus, long-term modifications of ME TRH-DE activity lead to limited changes in serum TSH concentration in adult rodents reared in standard conditions, possibly because of adaptations of TRH communication in the ME and/or anterior pituitary.

## 1. Introduction

During evolution, animals acquired efficient ways to sense and respond to internal changes, including selective and adaptive barriers. Circumventricular organs are medial brain structures open to the circulation that allow the transduction of peripheral signals to brain nuclei and, for secretory circumventricular organs, the release of efferent neuroendocrine messages [1]. At the base of the hypothalamus, a circumventricular organ, the ME has a ventricular wall made of β-tanycytes—elongated cells with an apical body exposed to the cerebrospinal fluid and a long basal process whose branches extend toward portal blood vessels [2]. ME (lateral β1 and central β2) tanycytes form barriers between the cerebrospinal fluid and the ME parenchyma, or between the ME itself and the arcuate nucleus (ARC) parenchyma [3] (Figure 1A). The ME is irrigated mostly by the superior hypophyseal arteries derived from the carotid artery, forming fenestrated capillaries in this region [4]. This anatomical configuration makes the ME sensitive to—and permissive to the passage of—peripheral metabolic signals such as thyroid hormones (TH), leptin, glucose, ghrelin, and other nutritional information into the third ventricle or nearby nuclei such as the ARC of the hypothalamus; some of this information is transduced to distant hypothalamic nuclei like the paraventricular nucleus of the hypothalamus (PVH) [5,6]. On the other hand, efferent signals generated by hypophysiotropic neurons are released into the proximity of fenestrated capillaries of the ME [7] and transported by portal vessels to the pituitary, where these signals influence the secretion of anterior pituitary hormones into peripheral circulation [8].

The control of the HPT axis by tanycytes illustrates the impact of ME cellular components over energy balance. Axons and terminal buttons from hypophysiotropic TRH (pGlu-His-Pro-NH2) neurons of the PVH are abundant in the external zone of the ME [9,10] (Figure 1A). Once released, TRH diffuses into the portal capillaries and is transported to the anterior pituitary where it stimulates the release of thyrotropin (TSH) into the general circulation and, consequently, TH secretion from the thyroid [11,12]. Circulating TH acts as a negative feedback signal at different levels of the HPT axis [13]. While 30% of TH secreted by the thyroid corresponds to 3,3′,5-triiodo-L-thyronine (T3), in addition, circulating thyroxine (T4) can be converted to T3 by tissue deiodinases; in β2-tanycytes, deiodinase type 2 (DIO2) produces T3 that might be taken up by nerve terminals of TRH-ergic hypophysiotropic neurons and retrogradely transported into the PVH to exert negative feedback control of *Trh* expression [11,14]. On the other hand, β2-tanycyte end-feet are in close contact with TRH terminals [15] and may also contribute to shaping TRH pulses since glutamate, co-released with TRH, induces endocannabinoid signaling by tanycytes that feeds back to reduce TRH release [16].

Tanycyte molecular machinery is remarkable for the abundance of TH transport, biotransformation, and sensing proteins (*Slc16a2*, *Slco1c1*, *Crym*, *Dio2*, *Thra*, and *Thrb*) and of the TRH-DE (EC 3.4.19.6)—a narrow-specificity omega ectopeptidase that, in the extracellular space, hydrolyses TRH to products inactive on the TRH receptors [15,17]. *Trhde* expression in β2-tanycytes could influence the bioavailability of TRH to the pituitary since the inhibition of ME TRH-DE activity enhances TRH recovery from ME explants; in addition, the peripheral injection of an inhibitor that reduces ME TRH-DE activity enhances the secretion of TSH induced by 1-hour cold stress in male rats reared in standard conditions [15]. Furthermore, the activity of TRH-DE in the ME is enhanced by TH, fasting, and TRH [15,17,18], suggesting it may adjust the TRH extracellular half-life. Finally, activation of the Gqα1 pathway in tanycytes—a pathway that couples to TRH receptor 1 (TRH-R1) [19,20]—increases the contact of tanycytes’ end-feet on portal vessels, while its inhibition enhances the secretion of TSH induced by chemogenetic stimulation of PVH TRH neurons [18]. These data suggest that either TRH-DE activity and/or end-feet positioning on portal capillaries is critical to adjust the output of TRH into the portal capillaries when TRH neurons are active. TRH-DE might thus be a target for the treatment of some pathologies derived from malfunction of the HPT axis [21], although the effects of a long-term perturbation of TRH-DE activity on HPT axis homeostasis are unknown.

To evaluate the impact of ME TRH-DE activity on basal HPT axis function in rodents housed in standard conditions, we compared the locus of expression of TRH-DE with that of TRH terminals in the ME of adult rats, determined the status of the HPT axis in adult mice homozygous or heterozygous for a global inactivation of *Trhde*, and analyzed the impact of the viral alteration of ME TRH-DE activity on HPT axis activity in adult rats. Rats and mice were used because some of our tools were not optimal for mice.

## 2. Materials and Methods

### 2.1. Animals

Rodents were bred at the institute’s animal house, a specific pathogen-free facility. They were kept under controlled temperature (21 ± 1 °C), humidity (30–55%), with lights on from 7:00 to 19:00 h, a standard diet (Teklad TD.2018SX chow [Envigo]), and purified water ad libitum.

B6;129S5-Trhde^tm1Lex^/Mmucd mice (RRID:MMRRC_032690-UCD) are a strain in which *Trhde* exon 1 (NCBI accession: NM_146241.1) is deleted and substituted by a β-galactosidase (β-gal) cassette by homologous recombination. The mice [22] were obtained from the Knockout Mouse Project, KOMP Repository, University of California, Davis (https://www.mmrrc.org (accessed on 27 January 2025). We back-crossed the original mutation to the C57BL/6NJ background, obtained from Jackson Laboratories, for up to 11 generations of mice, avoiding mixing parental lines or crossing relatives. C57BL/6NJ mice bearing both Trhde^tm1Lex^ alleles are abbreviated *Trhde* knock-out (KO) mice. Pups were genotyped, sexed after weaning, and separated by sex in different cages (2–4 per cage). Both male and female mice were sacrificed when approx. 2 months old, 3–7 h after switching to the light phase.

The Wistar rat colony was expanded from pairs of 3–4-month-old non-sibling male and female rats from Charles River (Wilmington, MA, USA), carefully recording crossings to keep each line independent and avoid inter-crossing. Pups were culled to 10 per mother to facilitate equal access to food and control their body weight. Animals were sexed after weaning to keep groups of 5 male rats per cage. Female rats were not used because data from mice did not reveal sex differences (see Section 3.2). Male rats were grouped in pairs starting at one month of age and were subjected to surgical or histological procedures when they reached 245–260 g body weight (approx. 2 months old) and sacrificed 2–3 weeks later, between 9 and 11 h. We did not use environmental enrichment but adult rats were maintained in cages with a high-lidded cover for climbing.

Whenever possible, experimental groups were formed by selecting each member from different mothers to limit siblings in a group. Animals were treated according to the *Guide for the Care and Use of Laboratory Animals (8th edition)*, National Research Council of the National Academies of the USA, and the Official Mexican Norm for the Production, Care, and Use of Laboratory Animals (NOM-062-ZOO-1999). Experiments and animal care were approved by the institute’s Biosafety (application number: 241010) and Bioethics (application number: 276) Committees. The number of animals used in each group and experiment is indicated in Appendix A.

### 2.2. Histochemistry for β-Gal in Mouse Brain Slices

Mice were injected i.p. with a lethal dose of pentobarbital sodium (210 mg/Kg, Sedalphorte, Salud y Bienestar Animal SA de CV, Mexico City, Mexico) and perfused first with 20 mL of PBS 0.01 M pH = 7.4 supplemented with 15,000 U/L heparin sulfate, followed by 30 mL of paraformaldehyde (PFA, 4% in PBS). Brains were excised and post-fixed with PFA at 4 °C for 6 h, then cryoprotected in sucrose solutions (from 20 to 30% until brain sinking), frozen with powdered dry ice, and stored at −80 °C until sectioning 25 µm slices. For detection of β-gal activity, slices were washed 3 times with buffer (MgCl_2_ 2 mM, sodium deoxycholate 0.01%, NP-40 0.02% in PBS 0.1 M, each for 10 min at room temperature) and then stained for at least 6 h with K_3_[Fe(CN)_6_] 5 mM, K_4_[Fe(CN)_6_] 5 mM, and X-gal 1 mg/mL [23]. Slices were post-fixed with PFA 4%, dehydrated, and mounted with DPX mounting medium on a slide. Standard bright light observation of blue precipitate was made with an Axioscop epifluorescence microscope equipped with Neofluar 20X/0.58 and 40X/0.75 (Zeiss, Oberkochen. Germany).

### 2.3. Immunofluorescence Detection of Vimentin and TRH-DE in Rats, and β-Gal and Vimentin in Mouse Brain Slices

Rodents were anesthetized with intraperitoneal (i.p.) pentobarbital sodium (rats: 120 mg/Kg) and perfused with PBS 0.01 M pH = 7.4 supplemented with 15 000 U/L heparin sulfate (50 mL for rats, 20 mL for mice) and then with PFA 4% (200 mL for rats, 50 mL for mice). Brains were processed, sliced, and stored as indicated above. Coronal sections of 25 μm (from bregma −1.4 to −3.6 mm in rats and bregma −1.58 to −2.3 mm in mice [24,25]) were made in a cryostat and cryoprotected in antifreeze solution (glycerol 30% and ethylene glycol 30% in PBS) at −20 °C until immunofluorescence processing. Sections were washed with PBS 0.01 M pH = 7.4 and then incubated with NaBH_4_ 0.1% for 15 min to remove autofluorescence, washed with PBS 0.01 M, and then blocked and permeabilized with a mixture of donkey normal serum (5%) and Triton X-100 (0.05%) in PBS 0.01 M for 3 h. Sections were incubated with primary antibodies (sources and dilutions listed in Appendix A), most of them diluted in PBS 0.01 M with Triton X-100 0.05% overnight, except for the TRH-DE antibody that was diluted in PBS 0.01 M. Sections were then incubated with a secondary antibody (Appendix A), finally contrasted with 4′,6-diamidino-2-phenylindole (DAPI), and mounted with Citifluor montage medium (EMS Acquisition Corp., Hatfield, PA, USA). Sections incubated with TRH-DE and β-gal antibodies were visualized with an Axioscop Zeiss epifluorescence microscope (magnifications with Zeiss Neofluar 20X/0.58, 40X/0.75) coupled to a ProgRes CF camera (Jenoptik Optical Systems, Jena, Germany), using Morpho Pro software 16.5.0 (Explora Nova, La Rochelle, France) with exposures ranging from 1000 to 4000 ms and gain: 1. For the vimentin extension analysis, mouse brain slices were observed with a confocal microscope (Olympus FV1000 Confocal Upright, Center Valley, PA, USA) using a UPLFL 40X/0.75, plus 3X magnification, at two laser wavelengths (405 and 635 nm), and with phase contrast to visualize capillaries. Nine confocal slices were stacked for analysis. The number of tanycyte vimentin-positive extensions in contact with basal capillaries was counted in 3 representative slices of the anterior, medial, and posterior regions of the ME.

### 2.4. Triple Immunofluorescence for Vimentin, TRH, and TRH-DE in Rat Brain Slices

The protocol was designed to maintain the structure of the extracellular domain of TRH-DE and TRH content in nerve terminals. Pore formation with saponin is subtler and more selective than permeabilization with detergents such as Triton X-100 or Tween-20 [26], which extract membrane proteins reducing immunodetection, a problem we initially faced when testing different TRH-DE antibodies. The chosen anti-TRH-DE antibody labeled rat brain areas in agreement with the known distribution of *Trhde* mRNA [27] as well as positive controls in rats (Appendix A).

Rats were anesthetized with pentobarbital sodium i.p. and perfused first with 50 mL of PBS 0.01 M pH = 7.4 supplemented with 15 000 U/L heparin sulfate, and then with 100 mL of a mix of PFA (3%) and acrolein (1%); excess acrolein was eliminated by perfusion with 50 mL of PFA (3%) [28]. Brains were removed and post-fixed with PFA (3%) at 4 °C for 6 h. Brains were processed, sliced, and stored as indicated above. Sections were incubated with NaBH_4_ 0.1% for 15 min, washed with PBS 0.01 M, and incubated with a mixture of 5% powdered milk (lactose-free Svelty) and saponin 0.1% in PBS 0.01 M for 3 h. Sections were incubated with antibodies (sources and dilutions listed in Appendix A) diluted in PBS 0.01 M pH = 7.4 with saponin 0.1%; first, with anti-TRH (at 4 °C overnight), the next day with a secondary anti-goat Cy3 (at room temperature for 2 h), then with anti-vimentin (at 4 °C overnight), and finally with a secondary anti-chicken Alexa-647 (at room temperature for 2 h). To reveal TRH-DE localization, slices were washed 4 times with PBS 0.01 M to remove any trace of saponin and then incubated (Appendix A) with anti-TRH-DE (at 4 °C overnight). Slices were then incubated with a secondary antibody Alexa-488 (at room temperature for 2 h), finally contrasted with DAPI, and mounted with Citifluor montage medium (EMS Acquisition Corp., Hatfield, PA, USA). Brain sections were visualized at the National Laboratory of Advanced Microscopy (UNAM) with a confocal microscope (Olympus FV1000 Confocal Upright) using UPLSAPO 10X/0.40, UPLFL 40X/0.75 and oil immersion UPLSAPO 60X/1.30, and laser wavelengths 405, 488, 543, and 635 nm.

### 2.5. Microinjection into the Rat Brain Third Ventricle (3V)

Rats were anesthetized with intramuscular ketamine (50 mg/Kg) and xylazine (10 mg/Kg) and placed in a stereotaxic apparatus (Kopf Instruments, Tigunga, CA, USA). A small hole was drilled through the skull; the injector (28 G internal cannula, Plastics One, Roanoke, VA, USA) coupled to a syringe pump (Bioanalytical Systems Inc., West Lafayette, IN, USA) was first inserted approximately 0.5 mm lateral to the middle axis to avoid bleeding due to damage of the central cerebral vein and then relocated to the center to push the vein. To identify the site of injection, a group of rats was injected with 10 µL of 0.04% of trypan blue in PBS (rate: 0.5 μL per min) in the third ventricle, in the proximity of the ME (0 mm from the midline, −2.4 mm anteroposterior from Bregma, and −9.4 mm dorsoventral from dura mater); we observed the stain in the third ventricle and ME; the injection site was confirmed by the gliosis that tracks it (Appendix A). Once the validity of the coordinates was established, rats were injected with 10 μL (0.5 μL/min) of adeno-associated virus (AAV) or sterile vehicle (0.001% Lutrol/F68 in PBS, Thermo Fisher Scientific, Logan, UT, USA). During surgery, body temperature was maintained using a far infrared warming pad. The cannula was slowly removed 10 min after the end of the injection. The hole in the skull was covered with bone wax, the skin was sutured, and rats were injected with 40,000 IU of benzylpenicillin (Benzetazil, Sandoz, CDMX, Mexico) intra-muscularly. Immediately after anesthetic recovery, pairs of animals were introduced into a clean, dry, and comfortable area, with some pellets on the bedding for food accessibility. Rats were maintained at two per cage, with food and water ad libitum, and body weight and food consumption were measured every week. In experiment 2, rats were habituated to tail blood extraction one week after AAV injection; then, at week 2, they were moved from their home cage to either an adjacent room at 4 °C (cold-stressed) or at 21 ± 1 °C (control temperature) at 9:00 h. Blood was collected from the tail in basal conditions and after 1 h of cold exposure. Blood samples were centrifuged to obtain serum for hormone determination.

### 2.6. Analysis of AAV Ability to Transduce Rat Tanycytes In Vivo

Two or three weeks after the 3V injection of either AAV4-GFP or self-complementary (sc) versions of AAV1-GFP, AAV2-GFP, AAV4-GFP, AAV5-GFP, AAV6-GFP, AAV8-GFP, AAV9-GFP, or AAVD/J-GFP (University of Iowa Viral Vector Core, Iowa City, IA, USA; Appendix A)—that expressed GFP under the cytomegalovirus promoter—rats were perfused with 50 mL of PBS 0.01 M/heparin 10 U/mL and 200 mL of 4% PFA in PBS. Cryopreservation, sectioning, and storage were as described above. To determine the distribution of GFP expression in the medio-basal hypothalamus, brain sections were mounted on a slide and some sections were processed for vimentin immunofluorescence to evaluate whether tanycytes were transduced. For the detection of GFP fluorescence, sections were mounted and directly visualized with an Axioscop Zeiss epifluorescence microscope (Zeiss Neofluar 20X/0.58) coupled to a Jenoptik camera using Morpho software (exposure of 4000 ms and gain: 1).

### 2.7. Efficiency of AAV1-TRH-DE and -TRH-DEt Vectors to Induce Changes in TRH-DE Activity in Cell Culture

To overexpress TRH-DE, or a truncated isoform of TRH-DE (TRH-DEt)—a dominant negative isoform [27]—we used recombinant AAV1 vectors that allowed the expression of TRH-DE (AAV1-TRH-DE) or TRH-DEt (AAV1-TRH-DEt) under the cytomegalovirus promoter (Cyagen Biosciences, Santa Clara, CA, USA). Efficiency of AAV1-TRH-DE and AAV1-TRH-DEt transgene expression was tested in COS-7 cells cultured as described [27]. A total of 3 × 10^5^ COS-7 cells were seeded and incubated with AAV1-TRH-DE at a multiplicity of infection (moi) of 1 × 10^4^ in 35 mm dishes. Activity of the dominant negative isoform transgene was evaluated by adding increasing proportions of AAV1-TRH-DEt to AAV1-TRH-DE. Cells were harvested after 48 h, the membranes extracted, and the TRH-DE activity was measured.

### 2.8. Overexpression of TRH-DE and TRH-DEt in Rats

To overexpress ME TRH-DE or TRH-DEt in vivo in the ME, 8.75 × 10^10^–4.37 × 10^11^ viral genomes (vg) of AAV1-TRH-DE or AAV1-TRH-DEt were injected into the third ventricle just above the ME (in 10 μL). Sham animals were injected with vehicle or 8.75 × 10^10^–4.37 × 10^11^ AAV1-GFP genomes. The number of viral genomes injected is indicated in the text and legends of the figures. In experiment 1, we injected 3V AAV1-TRH-DE or AAV1-TRH-DEt or AAV1-GFP, n = 5–6. In experiment 2, we injected AAV1-TRH-DE or AAV1-GFP, n = 3–6. In experiment 3, we injected AAV1-TRH-DE or vehicle, n = 2–4. In experiment 4, we injected AAV1-TRH-DEt or AAV1-GFP, n = 5. In experiment 5, we injected two increasing loads of AAV1-TRH-DE or AAV1-GFP, n = 6. In experiment 6, we injected AAV1-TRH-DE or TRH-DEt or AAV1-GFP, n = 7–8.

To overexpress TRH-DE or TRH-DEt in the supraoptic nucleus (SON) (experiment 7), we injected 4.37 × 10^9^ viral genomes of AAV1-TRH-DE or AAV1-TRH-DEt or AAV1-GFP in 0.5 μL at the base of each side of the brain (+/−2 mm from midline, −1.4 mm anteroposterior from Bregma, and −9.2 mm dorsoventral from dura mater), n = 5.

At either 2, 3, or 4 weeks after injection, rats were killed by decapitation by a trained technician, and trunk blood serum was collected and stored at −70 °C. White adipose tissue (WAT) was weighted. Tissues were stored at −70 °C for measurement of TRH-DE activity. See also Appendix A.

### 2.9. Hormone Assays

For rats, serum TSH (IB79181, IBL-America, Minneapolis, MN, USA), as well as total T3 (tT3) and total T4 (tT4) (Diagnóstica Internacional, Zapopan, México), arginine-vasopressin (AVP) (OKEH03099, Aviva Systems Biology, San Diego, CA, USA), prolactin (PRL) (80560, Crystal Chem, Elk Grove Village, IL, USA), growth hormone (GH) (80586, Crystal Chem), and interleukin-6 (IL-6) (ab119548, Abcam, Cambridge, UK) concentrations were measured with ELISA kits by duplicate within the same assay. The sensitivity ranges were as follows: 0.1–80 ng/mL for TSH; 0.1–10 ng/mL for tT3; 0.1–25 μg/dL for tT4; 31.3–2000 pg/mL for AVP; 5–80 ng/mL for PRL; 0.025–1.5 ng/mL for GH; and 31–2000 pg/mL for IL-6. To ensure a linear dose–signal relationship in the tT3 and tT4 assays, serum from hypothyroid rats was added to the standard curve in accordance with Bianco et al., 2014 [29]. Serum TSH concentration was also measured with a radioimmunoassay (RIA) using rNIDDK reagents (Bethesda, MD, USA); detection limit: 25 ng/mL; intra-assay and inter-assay variation coefficients < 10%. To quantify serum TRH concentration, orthophenantroline 1 mM final—an inhibitor of thyroliberinase—was added to blood at recollection, and serum was kept at −20 °C. Serum was thawed and mixed with 4 volumes of cold acetic acid 20% and centrifuged at 12,000× *g* for 30 min at 4 °C; 2 volumes of cold methanol were added to the supernatant, which was centrifuged; and the supernatant was evaporated. Residue was resuspended in RIA buffer and the TRH was quantified by RIA using the R2 antibody [30]. The sensitivity range was 25–2500 pg. Intra- and inter-assay variation coefficients were <10%. Serum TSH, tT3, and tT4 were also quantified with a Milliplex MAP Rat Thyroid Panel (RTHYMAG-30K, Millipore, Billerica, MA, USA), which uses antibody-coated magnetic beads coded with fluorescence to simultaneously measure multiple analytes in a sample. The methods used to measure serum TSH concentration are indicated in the legends.

For mice, serum leptin and tT4 concentrations were determined with ELISA kits (Crystal Chem and Diagnostica Internacional, respectively); corticosterone and TSH concentrations by RIA [31] within the same assay. The sensitivities were as follows: 0.1 ng/mL for serum leptin; 50 ng/mL for serum tT4; 1 ng/mL for serum tT3; 25 ng/mL for serum corticosterone; and 0.2 ng/mL for serum TSH.

### 2.10. TRH-DE, Thyroliberinase, and Aminopeptidase N (APN) Activities

The TRH-DE activity assay was based on the protocol described by [32], with some modifications. Brains were extracted and washed with fresh PBS. The ME, a small protuberance in the basal hypothalamus, was carefully dissected using 2 mm Vannas Spring Scissors (15000-03; Fine Science Tools, Foster City, CA, USA) and immediately frozen in dry ice. Tissues were sonicated in 50 mM sodium phosphate buffer (pH = 7.4) and centrifuged at 80,000 rpm. After a wash with phosphate buffer, pellets were sonicated again to obtain membrane extracts and stored at −70 °C until the TRH-DE activity was measured. For serum thyroliberinase activity, 5 μL of blood serum was used per assay. Membranes or serum were added to a mix of bacitracin and N-ethyl-maleimide and a saturating amount of dipeptidyl peptidase IV. After 10 min of preincubation, TRH–methyl coumarin (TRH-MCA), a fluorogenic substrate of TRH-DE, was added to the reaction tubes. Once His-Pro-MCA is released by TRH-DE activity, dipeptidyl peptidase IV releases MCA. An aliquot of the reaction was taken and mixed with the same amount of cold methanol every hour. The last reaction sample was taken after 3 h. MCA fluorescence was measured with a Nanodrop 3300 fluorospectrometer (Thermo Fisher Scientific) excited with a UV LED (400–750 nm) and read at 437 nm. For APN, activity was assayed with 400 µM Ala-β-naphtylamide (βNA) as the substrate in 100 mM Tris-HCl pH 7.5. Released βNA was determined in a fluorometer (excitation, 335 nm; emission, 410 nm). Enzymatic preincubations and assays were performed at 37 °C; assays were performed under initial velocity conditions in duplicate.

### 2.11. Genomic DNA PCR

To determine the number of Trhde^tm1Lex^ alleles, DNA was extracted from the tails of mice, purified as described (basic protocol 1; [33]), and amplified by PCR using the oligonucleotides described in Appendix A. The products were separated by gel electrophoresis and detected by staining with ethidium bromide. WT mice are characterized by one 421-base-pair band, and KO mice by one 494-base-pair band [22] (https://www.mmrrc.org; accessed on 27 January 2025).

### 2.12. End-Point RT-PCR

Gene expression in the mouse tissues was evaluated by end-point RT-PCR. After dissection, tissue was frozen onto dry ice. Total RNA, isolated as described [34], was quantified by spectroscopy, the integrity was examined by gel electrophoresis, and staining was performed with ethidium bromide. *Trh*, *Trhde*, and *Dio1* mRNA levels were determined using *Actb* or *Hprt* or *Ppia* (indicated in Figure 2’s legend) mRNA as internal control. The oligonucleotides used are described in Appendix A.

Briefly, RNA (1 μg) was reverse-transcribed; amplification conditions were such that there was linearity between the number of cycles and the quantity of DNA products. For each cDNA product, aliquots of the amplicons were mixed and electrophoresed together; the amplicons were identified according to molecular weight and quantified by densitometry. The ratio of *Trh* or *Trhde* or *Dio1* over the internal control amplicon signal was computed. Genotypes did not change *Actb* or *Hprt* amplicon levels.

### 2.13. RT-qPCR

Gene expression in rat tissues was evaluated by RT-qPCR. For PVH mRNA analysis, frozen brains were mounted in the cryostat and a 200 µm slice centered on the PVH region (−0.06 mm to Bregma, according to [24]) was cut. Bilateral PVH samples were obtained with a 1-millimeter internal diameter punch, and RNA was extracted using an RNAEasy Plus Micro kit (74034, Qiagen, Hilden, Germany) using provider protocol. For anterior pituitary analysis, frozen tissue RNA was extracted with 500 uL of QIAzol Lysis Reagent (79306, Qiagen) using provider protocol. Genomic DNA was removed with a RapidOut DNA Removal Kit (K2981, Thermo Fisher Scientific). RNA concentration was measured with a Nanodrop 2000 (Thermo Fisher Scientific). cDNAs were generated with 0.5 ug of RNA using random primers (Thermo Fisher Scientific).

qPCR primers are described in Appendix A; *Rplp0* probes were used for normalization since *Rplp0* expression was independent of the treatments. For the PVH, the qPCR reaction was performed using Kapa Sybr^®^ fast (Roche). For the anterior pituitary, qPCR reaction was performed using qPCRBIO SyGreen Mix (PB20.17, PCR Biosystems, Wayne, PA, USA). A Qiagen Rotor Gene Q thermocycler (Qiagen) was used. A posterior melt analysis was made to verify the absence of non-specific PCR products.

### 2.14. Data Presentation and Statistical Analyses

The animals were excluded from analysis if they showed low activity and appetite, loss of more than 20% of their body weight, or piloerection changes in respiratory pattern. We did not exclude data points. Results are presented as mean ± SEM. The number of independent determinations is included in figure and Supplementary figure legends, Supplementary tables, and in the text. For mice, data were analyzed using GraphPad Prism version 9.5.1. For rats, data were processed and analyzed using R Statistical Software (v. 4.2.2) [35]. Graphs were made using R libraries cowplot (v. 1.1.1) [36], and tidyverse packages ggplot2 (v. 3.3.6), dplyr (v. 1.09), tidyr (v. 1.2.0), and stringr (v. 1.4.0) [37]. The statistical tests used are described in the legends; the statistical parameters as well as the results are shown in Appendix A.

To identify correlations between variables from different experiments, data from rats were standardized using a z-value transformation, and ME TRH-DE activity was transformed using its natural logarithm as ME TRH-DE expression showed skewness due to the high TRH-DE activity in the ME of several AAV1-TRH-DE-injected animals. For this, the mean and standard deviation for each variable were calculated for the control group (vehicle or AAV1-GFP). Each z-value was obtained by subtracting the mean of the control group from the original value and dividing by the standard deviation of the control group. Data from experiments 3, 4, 5, 6, and 7 were grouped and correlation coefficients were determined using Pearson’s, Spearman’s, and Kendall’s methods. *p* values were obtained using standard methods from base R libraries (*t*-test, algorithm AS 89, and rho test, respectively).

## 3. Results

### 3.1. Localization of TRH-DE Immunoreactivity in the ME of the Adult Rat

In rats, ME TRH-DE activity was consistently detected at postnatal day (PND) 10, reaching a maximum value at PND 60 (Appendix A), a time-point chosen for all additional experiments. TRH-DE-specific activity was much higher in the ME than in the anterior or posterior pituitary or whole brain, as previously reported [15,17,27,38]. A substantial TRH-DE activity per microliter of serum—corresponding to thyroliberinase, an isoform of TRH-DE produced by the liver [39]—was also detected (Appendix A), in agreement with previous data [38].

Although *Trhde* mRNA expression is prominent in the rat ME in tanycyte vimentin-positive domains that extend down to the ME external zone [15], the spatial relationship between TRH-DE protein, tanycyte end-feet, and TRH terminals is unknown. Triple labeling immunofluorescence experiments showed the distribution of TRH-DE, vimentin, and TRH in the ME and neighboring region of the male rat. In the hypothalamus, vimentin staining was enriched along the ventro-lateral and ventral walls of the third ventricle, overlapping ependymocyte and tanycyte domains (Figure 1B). In the ME, β2-tanycyte bodies with visible nuclei packed in the ependymal layer contacting the third ventricle extend a vimentin process in the direction of the external zone where it subdivides among TRH axons and terminal buttons [15]. A TRH-DE immunofluorescence signal was detected in limited regions of the medio-basal hypothalamus, enveloping ARC cells and specific regions of the ME (Figure 1B,C). Alpha-tanycytes were not labeled with the TRH-DE antibody, whereas labeling was predominant in the medial/external zone of the ME, surrounding vimentin fibers, and in close relationship with the TRH terminal buttons, outlining the domains of ventral fenestrated capillaries; however, TRH and TRH-DE signals were not colocalized (Figure 1D,E). Other TRH-DE-positive structures in the ME were pars tuberalis cells (Figure 1D). This pattern of TRH-DE immunodetection was consistent along the rostro-caudal extent of the ME (Appendix A). These observations show that TRH-DE is close to the putative sites of TRH release in the rat ME and could, therefore, influence the amount of TRH reaching the portal vessels. Hence, we measured the impact of the long-term modification of TRH-DE activity over serum TSH concentration and other HPT axis parameters in rodents housed in standard conditions.

### 3.2. Effect of Global Elimination of Trhde Expression on Basal HPT Axis Activity in Mice

*Trhde* KO mice (in the B6129S5 background) have a normal phenotype, except that body weight is slightly lower than in wild-type (WT) or heterozygous (HT) mice [22]; the status of the HPT axis has not been reported (https://www.mmrrc.org; accessed on 27 January 2025). To test whether the absence of TRH-DE activity influences HPT axis activity, we back-crossed the original mutation to the C57BL/6NJ background for up to 11 generations of mice. Mice were fed and reared in standard conditions. HT mice fertility was normal. Both male and female mice were tested after 1, 5, 9, and 11 back-crossings when two months old. All mice appeared healthy; at 6 months of age, survival was 100% (n = 36).

The β-gal reporter was used to compare the expression sites of *Trhde* in transgenic male mice with those in WT mice (Allen mouse brain atlas [40]). Staining was found in *Trhde* HT and KO mice but not in WT mice, restricted to previously reported loci of expression: high intensity in the hippocampus and cerebral cortex; intermediate in the amygdala, preoptic area of the hypothalamus, or the PVH. In contrast to the detection of *Trhde* expression and TRH-DE activity in the ME of WT mice (Allen mouse brain atlas [40], and Figure 2C), β-gal activity was not detected along the ventro-lateral walls of the third ventricle and in the ME (Appendix A), even with longer reaction times than indicated in the Materials and Methods Section. β-gal was detected by immunohistochemistry in *Trhde* KO mice in the same regions as the X-gal hydrolysis product, including ARC and pars tuberalis cells, although the signal was scarce in the ependymal layer corresponding to ME tanycytes (Appendix A). Failure to detect activity or immunoreactivity of β-gal along the base of the 3V is not due to the inability of tanycytes to express β-gal as activity is detected in tanycytes in other transgenic models [41]. The specific insertion locus can influence the activity of LacZ reporters in mice, and detection can diminish in advanced postnatal stages, explaining cellular differences in the detection of β-gal in multiple models [42,43]. Since TRH-DE activity is almost eliminated in the ME of *Trhde* KO mice (Figure 2C), it is likely that the locus of insertion in transgenic *Trhde* mice impeached expression of β-gal in tanycytes.

The following data were independent of the number of generations. In male mice, *Trhde* expression in the brain and pituitary, and ME TRH-DE and thyroliberinase-specific activities were reduced in *Trhde* HT and almost eliminated in *Trhde* KO compared to WT mice (Figure 2A–D). After an i.p. injection of TRH, the concentration of TRH detected in the serum was lower in WT than in *Trhde* HT and KO animals (Appendix A), confirming that TRH-DE contributes to TRH elimination in blood [44]. However, besides the notable differences in TRH-DE activity, we did not find other genotype-dependent differences in most parameters. *Trhde* KO mice had a body weight that was not different from that of WT mice (Figure 2E). Brain APN activity, the closest member of the M1 family of metalloaminopeptidases (Appendix A), food and water intake (Figure 2F,G), circulating leptin and corticosterone concentrations (Figure 2H,I), PVH *Trh* mRNA levels (Figure 2J), serum TSH and tT4 concentrations (Figure 2K,L), and *Dio1* mRNA levels in the liver (Figure 2M) were not different between WT and KO mice. Notwithstanding, the number of vimentin filaments at the base of the external zone of the ME in contact with capillaries per area was higher in *Trhde* KO male mice than in WT animals (Figure 2N,O). For female mice, although we did not analyze some parameters, the data obtained were like those in male mice (Figure 2; Appendix A).

These results suggest that TRH-DE has no influence on HPT axis activity in standard conditions, or that the effect of the *Trhde* gene mutation is compensated by adjustments of HPT control elements during development, such as morphological alterations of the ME. To test the specific role of ME TRH-DE in adult animals, we used a viral vector approach in rats, which a wealth of histologic and physiologic data place as a mammalian model of choice to study the HPT axis [29], and for which access to any brain region for in vivo cannulation is relatively straightforward compared to other small vertebrates [45].

### 3.3. Identification of AAV Serotypes Whose Injection into Rat 3V Transduces ME Tanycytes

An increasing number of experimental applications of AAVs demonstrate consistent effects in transduced host cells, including short- and long-term expression in vivo. However, brain manipulation with AAVs to study energy balance is limited to mouse models [46]. We tested the effect of delivery in the 3V of adult rats of different AAV serotypes expressing GFP in a conventional AAV construction (Appendix A) or in a self-complementary construction to enhance the visualization of the transduced cells. As expected, two or three weeks after the viral injection, patterns of GFP expression were serotype-dependent, varying in GFP abundance and proportion of cell types, at identical multiplicity of infection.

Since β2-tanycytes are the cells that most likely contribute to *Trhde* expression and activity in the ME ([15,27,47] and this study), we initially focused on the expression of GFP in the ME and surrounding areas. While scAAV1-GFP induced a consistent transduction of ME tanycytes (Appendix A), scAAV2-GFP transduced some ME tanycytes and mostly other cells (Appendix A), and AAV4-GFP transduced varicosities or nerve terminal buttons present in the external layer of the ME (Appendix A); for other serotypes, the level of transduction in the ME was low or undetectable (Appendix A). At the multiplicity level tested (4.5 × 10^9^ viral genomes), injection of scAAV1-GFP resulted in the highest GFP signal in tanycytes (Appendix A). Although we have not quantified the proportion of β2-tanycytes that are transduced with scAAV1-GFP, qualitative observations suggest that it is in the 20–50% range. Other brain regions were also transduced by 3V AAV, dependent on the serotype.

The choroid plexus and subcommissural organ were labeled by AAV4, AAV5, and AAV6 (Appendix A) but not by AAV1 or other serotypes. Except for AAV5, meningeal cells were transduced by all serotypes (Appendix A). Although for most AAV serotypes (except AAV4 and AAV5) we observed GFP fluorescence in varicosities and compartments resembling Herring bodies in the posterior pituitary, we did not observe fluorescence in the anterior pituitary (Appendix A). Posterior pituitary transduction coincided with labeling of the cell soma in the SON when AAV1, AAV2, AAV6, and AAV8 were injected (Appendix A). We did not detect labeling of the PVH, albeit other hypothalamic nuclei were labeled by AAV2, AAV8, AAV9, and AAVD/J (Appendix A).

Based on the cellular specificity and intensity of GFP transduction, we selected the AAV1 serotype to manipulate TRH-DE activity in the ME. Although the specificity of AAV1 for the ME was not absolute, key hypothalamic centers in energy balance (ARC, dorso- and ventro-medial hypothalamus, and PVH) did not express GFP (Appendix A). As expected, after stereotaxic surgery, body weight was lower than in intact animals for 1 month; however, compared to vehicle-injected animals, AAV1-GFP administration neither changed the rate of body weight increase (Appendix A), nor modified basal serum TSH concentration (Appendix A), nor induced a detectable concentration of IL-6 in serum (less than 31 pg/mL, n = 3).

These data suggest that AAV1 delivery into the 3V of adult rats allows for the relatively specific modification of gene expression in ME cells without modifying basal HPT axis activity, and thus seems adequate to explore the effect of modifying TRH-DE activity on HPT axis variables.

### 3.4. Temporal Dynamics of over- or Under-Expression of TRH-DE Induced by AAV1 Constructions and Influence on TSH Secretion and HPT Axis Parameters

AAV1 vectors were designed to overexpress TRH-DE (AAV1-TRH-DE) or TRH-DEt (AAV1-TRH-DEt), a truncated isoform of TRH-DE with dominant negative influence [48] (Figure 3A). Both AAV vectors transduced COS-7 cell cultures efficiently and either increased (AAV1-TRH-DE) or reduced TRH-DE activity (AAV1-TRH-DEt) in a concentration-dependent manner (Appendix A). TRH-DE immunoreactivity in the external zone of the ME was enhanced in animals injected with AAV1-TRH-DE or AAV1-TRH-DEt when compared with AAV1-GFP injection in the 3V (experiment 1, Appendix A). Our success rate at modifying ME TRH-DE activity in either direction was variable; we detected changes in TRH-DE ME activity of at least 50% compared to the mean of control groups in 65 ± 6% (n = 4) of AAV1-TRH-DE injections and 55 ± 5% (n = 2) with AAV1-TRH-DEt injections.

Regarding the cellular specificity of the expression of TRH-DE or TRH-DEt, based on AAV1-GFP transduction data, an effective AAV1-TRH-DE injection should have raised TRH-DE activity in tanycytes and other ME cell types, as well as in SON magnocellular neurons. In contrast, while AAV1-TRH-DEt injection should induce TRH-DEt expression in the same cells, the dominant negative catalytic effect of this truncated isoform of TRH-DE should be limited to cells that express the complete TRH-DE isoform and be specific for TRH-DE [48]. Because *Trhde* expression in the medio-basal hypothalamus is predominant in β-tanycytes [15,17,47], TRH-DEt injection into the 3V should produce an almost specific knock-down of β-tanycyte TRH-DE activity.

Exposure of rats to 4 °C for up to 2 h transiently increases *Trh* expression in the PVH and serum TSH concentration [49,50]. Although, in experiment 2, injection of AAV1-TRH-DE did not change serum concentrations of TSH 2 weeks later in rats maintained at room temperature (AAV1-GFP: 331 ± 126 (n = 3); AAV1-TRH-DE: 499 ± 35 pg/mL (n = 5); *p* = 0.157, Student’s *t*-test), tT3 (AAV1-GFP: 2018 ± 322 (n = 3); AAV1-TRH-DE: 2914 ± 279 pg/mL (n = 5); *p* = 0.088, Student’s *t*-test), and tT4 (AAV1-GFP: 83 ± 3.9 (n = 3); AAV1-TRH-DE: 89 ± 1.6 ng/mL (n = 5); *p* = 0.1568, Student’s *t*-test), a rise in serum TSH concentration in AAV1-GFP rats was detectable after 1-hour cold exposure, a response blunted in AAV1-TRH-DE animals (Figure 3B). These data demonstrate that TRH-DE activity in the ME controls TSH secretion when a stimulus induces TRH release, as proposed previously [15], and suggest that ME TRH-DE activity does not control the basal serum concentration of TSH in male rats. To test this latter suggestion, we performed other experiments.

In experiment 3, compared with the vehicle group, we detected a rise in ME TRH-DE activity two weeks after the intra-3V delivery of AAV1-TRH-DE, an increase that was reversed at 3 weeks (Figure 3C). The mechanistic basis of this result is unknown; it contrasts with data that show that AAV transduction generates stable long-term expression of proteins in mouse tanycytes [51]. The regional specificity of the transduction was confirmed as we did not detect changes in TRH-DE activity in the anterior pituitary (Appendix A) or in serum thyroliberinase activity 2 or 3 weeks after AAV1-TRH-DE delivery in the 3V (Appendix A). The AAV1-TRH-DE injection produced a small but significant decrease in serum TSH concentration (Figure 3D) 3 weeks after viral delivery; serum concentrations of total TH (Appendix A), of GH (AAV1-GFP: 34.3 ± 18.5 (n = 2); AAV1-TRH-DE: 22.9 ± 12.3 ng/mL (n = 4); *p* = 0.621, Student’s *t*-test), and body weight gain (Appendix A) were not altered.

Because of this unexpected time-dependent reversal of the 3V AAV1-TRH-DE effect on ME TRH-DE activity, we restricted the evaluation of transduction effects to the second week post-viral delivery for the next experiments. In experiment 4, compared to AAV1-GFP-injected animals, TRH-DE activity decreased in the ME of some animals injected with AAV1-TRH-DEt (Figure 3G), while thyroliberinase activity was unchanged (Appendix A). AAV1-TRH-DEt injection into the 3V enhanced serum TSH concentration (Figure 3H) but neither changed serum TH concentration (Figure 3I,J) nor body weight gain (Appendix A).

In experiment 5, we increased the quantity of AAV1-TRH-DE and AAV1-GFP injected. Compared to the average activity of the AAV1-GFP group, AAV1-TRH-DE injection increased ME TRH-DE activity 8-fold (Figure 4A). Nevertheless, anterior pituitary *Dio2*, *Thrb*, and *Tshb* expression (Figure 4C,D,F), serum TSH concentration (measured with either RIA or ELISA; Figure 4B, Appendix A), and serum concentration of tT3 (AAV1-GFP: 72.8 ± 8.0 (n = 10); AAV1-TRH-DE: 61.6 ± 3.9 ng/mL (n = 10); *p* = 0.22, Student’s *t*-test) or tT4 (AAV1-GFP: 3.28 ± 0.31 (n = 10); AAV1-TRH-DE: 3.60 ± 0.37 μg/dL (n = 10); *p* = 0.5047, Student’s *t*-test) did not change. In contrast, AAV1-TRH-DE injection increased anterior pituitary *Trhr* and *Prl* expression (Figure 4E,G), two genes that are regulated by TRH [51,52,53].

Although in experiment 1, intracerebroventricular delivery of AAV1-TRH-DE or AAV1-TRH-DEt changed TRH-DE ME activity in the expected direction (AAV1-GFP: 1.00 ± 0.11 (n = 5); AAV1-TRH-DE: 2.53 ± 0.92 (n = 4); AAV1-TRH-DEt: 0.88 ± 0.07 pmoles/min × ME (n = 5); *p* = 0.058, one-way ANOVA), it did not change *Trh* PVH expression compared to AAV1-GFP (Appendix A), or serum TSH concentration (AAV1-GFP: 1.40 ± 0.63 (n = 5); AAV1-TRH-DE: 0.86 ± 0.34 (n = 4); AA1-TRH-DEt: 0.88 ± 0.24 pmoles/min × ME (n = 5); *p* = 0.479, one-way ANOVA).

In experiment 6, stereotaxic injections of AAV1 vectors induced subtle effects in ME TRH-DE activity (AAV1-GFP: 1.32 ± 0.11; AAV1-TRH-DE: 1.53 ± 0.19; AAV1-TRH-DEt: 0.88 ± 0.08 pmoles/min × ME (n = 7); difference between AAV1-TRH-DE and AAV1-TRH-DEt: *p* = 0.008, one-way ANOVA), with no effect over serum concentrations of TSH (AAV1-GFP: 1.74 ± 0.37; AAV1-TRH-DE: 1.87 ± 0.44; AAV1-TRH-DEt: 1.57 ± 0.36 ng/mL (n = 7); *p* = 0.859, one-way ANOVA), tT3 (AAV1-GFP: 67.4 ± 24.0; AAV1-TRH-DE: 108 ± 22.3; AAV1-TRH-DEt: 87.5 ± 25.1 ng/mL (n = 7); *p* = 0.486, one-way ANOVA), tT4 (AAV1-GFP: 6.29 ± 0.32; AAV1-TRH-DE: 5.88 ± 0.23; AAV1-TRH-DEt: 5.32 ± 0.37 μg/dL (n = 7); *p* = 0.120, one-way ANOVA), or leptin concentration, white adipose weight, body weight (not shown), or serum triglyceride concentration (Appendix A).

In summary, the experiments in which we intended to modify TRH-DE expression in the ME did not show a reproducible effect on basal serum TSH and TH concentrations. This may be the result of insufficient statistical power, and/or of variable efficacy and precision of the stereotaxic AAV injection, a parameter which could not be measured directly except for its effect on ME TRH-DE activity. To consider some of these difficulties, we analyzed the correlations between ME TRH-DE activity and multiple HPT axis parameters in each experimental condition.

### 3.5. TRH-DE Activity in the ME and Basal Serum TSH Concentration Are Negatively Correlated in Adult Male Rats

The output of multiple experiments was grouped to analyze correlations between variables. As expected, serum leptin concentration strongly correlated positively with white adipose weight in all the experimental groups (Appendix A). ME TRH-DE activity was negatively correlated with serum TSH concentration in AAV1-GFP-treated animals, or when considering all animals (significant Pearson’s, Spearman’s, and Kendall’s correlations) (Figure 5A), and with serum concentration of tT3 (Figure 5B) but not with serum concentration of tT4 (Figure 5C). ME-TRH-DE and thyroliberinase activities did not correlate (Figure 5D) but we observed robust and significant correlations (Pearson’s, Spearman’s, and Kendall’s tests) between thyroliberinase activity and serum TSH concentration, with opposite directions in the AAV1-TRH-DE and AAV1-TRH-DEt groups (Figure 5E), making it unlikely that thyroliberinase activity sets serum TSH concentration. Thyroliberinase activity correlated negatively with serum tT3 concentration in the AAV1-GFP group (Figure 5F) but not with serum tT4 concentration (Figure 5G). As for anterior pituitary transcripts, data gathered from experiments 1 and 5 show that ME TRH-DE activity correlated negatively with *Prl* expression in the AAV1-GFP group, as with serum PRL concentration (Appendix A), and serum tT4 concentration was negatively correlated with *Dio2*, *Thrb*, and *Prl* expression in the same group (Appendix A).

To identify other factors affecting the correlation between ME TRH-DE activity and serum TSH concentration, partial and semi-partial correlations were calculated [54]. The partial correlation between ME TRH-DE activity and serum TSH concentration was still significant in the global data (r = −0.307, *p* = 0.015) and AAV1-GFP-treated animals (r = −0.49, *p* = 0.016) when thyroliberinase activity was maintained constant. When the other factors (tT3, tT4, adipose tissue weight, and body weight gain) were compensated for, the global correlation was still significant (r = −0.32, *p* = 0.019), showing that none of the factors tested contributed to the negative correlation between ME TRH-DE activity and serum TSH concentration.

### 3.6. Exploring the Influence of Posterior Pituitary TRH-DE Activity on Basal Serum TSH Concentration in Rats

The 3V injection of scAAV1-GFP strongly enhanced GFP expression in the posterior pituitary (Appendix A). This result was consistent with experiment 3, in which rats that received a 3V injection of AAV1-TRH-DE had a higher posterior pituitary TRH-DE activity than vehicle-treated rats. As in the ME, this change in posterior pituitary activity was detected at 2 weeks but not at 3 weeks (Appendix A).

TRH concentration (but not *Trh* expression), and TRH-DE-specific activity (but not *Trhde* expression) are higher in the rat posterior pituitary than in the anterior pituitary, although lower than in the ME [27,28,55] (Appendix A for TRH-DE activity). *Trh* may be expressed by magnocellular neurons of the paraventricular and/or the SON of the hypothalamus [56], and TRH may control the release of vasopressin and oxytocin [57]. Regarding posterior pituitary TRH-DE activity, it might localize to magnocellular neuron axons as significant expression of *Trhde* was detected in the SON of wild-type rats by transcriptomic profiling [58]. Small amounts of *Trhr* (and *Trhde*) are expressed by pituicytes in mice [59], suggesting that TRH might have a local role. We analyzed the functional impact of this locus of TRH-DE activity on serum TSH concentration.

To avoid a direct interference of AAV injection on PVH TRH neurons and tanycytes, in experiment 7 we injected AAV1-TRH-DE into both sides of the SON of the rat to induce an overexpression of TRH-DE in the posterior pituitary and measure its possible effects on serum TSH concentration. AAV1-TRH-DE injection into the SON successfully induced high TRH-DE activity in the posterior pituitary (Figure 6C), a small increase in the ME (Figure 6A) consistent with the passage of SON axons through the ME, and no changes in the anterior pituitary (Figure 6B) when compared to the AAV1-GFP group. Despite these marked changes in TRH-DE activity in the posterior pituitary, we did not detect a significant change in the serum concentration of TSH (Figure 6D), tT3 (AAV1-GFP: 60.6 ± 4.4; AAV1-TRH-DE: 52.7 ± 8.1 ng/mL (n = 5); *p* = 0.282, Student’s *t*-test), or tT4 (AAV1-GFP: 2.52 ± 0.36; AAV1-TRH-DE: 3.21 ± 0.47 μg/dL (n = 5); *p* = 0.1156, Student’s *t*-test).

## 4. Discussion

The amount of TRH released by hypophysiotropic neurons that reaches the portal capillaries is not only controlled by their spiking patterns but also by events that operate at nerve terminals and in β2-tanycytes. The relative physiological relevance of the phenomena occurring in the ME has yet to be fully clarified, including the specific contribution of β2-tanycyte TRH-DE. In this report, we show that TRH-DE is close to the putative sites of TRH release in the rat ME, that global *Trhde* KO mice do not show alterations in basal HPT axis parameters, and that long-term modification of TRH-DE activity in the ME of adult male rats has a small effect on basal serum TSH concentration, in contrast to a stronger one when hypophysiotropic TRH neurons are activated by 1-h cold exposure.

### 4.1. TRH-DE Is Close to the Putative Sites of TRH Release in the Rat ME

*Trhde* mRNA is detected in tanycyte cell bodies and in their cytoplasmic process [15]. We demonstrated that TRH-DE immunolabeling is not homogeneous along the tanycyte cell surface in the ME, being more prominent in the intermedial and external zones—near the portal capillaries and TRH terminals—than in other subcellular loci. The localization of *Trhde* mRNA into the cytoplasmic extensions may contribute to the polarization of TRH-DE localization since polarization of mRNAs and local translation enrich specific proteins at defined domains of the cell [60]. Although factors contributing to the localization of TRH-DE in tanycyte end-feet are unknown, they probably follow mechanisms identified for other proteins in various glial cell types [61].

Thus, ME TRH-DE activity, previously detected in the whole ME [15], is likely concentrated in the intermedial/external layer of the ME, where it seems ideally suited to control TRH output into the portal capillaries. This subcellular information, together with a previous report showing that TRH-DE inhibition enhances the recovery of TRH released from ME explants [15], supports a critical role of tanycyte TRH-DE activity in the control of TRH flux into portal vessels. This hypothesis was further tested with models in which a long-term modification of TRH-DE activity was established.

### 4.2. Two Models for Long-Term Modification of TRH-DE Activity

Young adult global KO *Trhde* mice maintained in standard conditions, which are almost completely devoid of TRH-DE activity, either in the ME or in other sites of *Trhde* expression, are apparently healthy, fertile, with no obvious phenotypic differences with WT animals, except for a small decrease in body weight in some cohorts (see also https://www.mmrrc.org; accessed on 27 January 2025; [22]). Although this model should give insights into HPT axis control by TRH-DE, it has limitations. To eliminate the potential developmental effects of *Trhde* KO and to focus on the specific role of ME TRH-DE activity, we also used a model in which we manipulated adult male rat ME *Trhde* expression with AAV.

Transduction of different brain cell types mediated by viral vectors has been extensively documented, although information about the interactions of AAVs with different subtypes of glial cells is still incomplete, especially in rats [46,62]. We tested the efficiency and specificity of different AAV serotypes injected into the 3V to transduce cells of the rat ME; 3V scAAV1 delivery targeted β2-tanycytes, among the occasional targeting of round-shaped cells in the ME. These findings are consistent with previous experiments in mice, in which tanycyte transduction was induced with a viral construction of mosaic AAV1 and AAV2 capsids injected into the lateral ventricles [18]. However, the injection of AAV1 vectors into the rat 3V also led to labeling of the posterior pituitary, which could be due to the diffusion of viral particles to ventral brain areas, including SON magnocellular neurons sending axons into the posterior pituitary. Thus, to induce a sustained and mainly local modification of TRH-DE activity in the ME, we used two vectors: AAV1-TRH-DE to increase and AAV1-TRH-DEt to decrease ME TRH-DE activity.

### 4.3. ME TRH-DE Controls the Phasic Output of TSH

One of the best-known stimuli that activates the HPT axis is cold (4–5 °C) exposure; in male rats reared in standard conditions, it rapidly enhances TRH mRNA levels in PVH neurons, TRH release into blood, and serum TSH concentration [49,50,63,64,65,66]. Previous reports suggested that tanycyte TRH-DE activity controls serum TSH concentration and/or HPT axis activity when a phasic induction of TRH release from the hypophysiotropic PVH TRH neurons is promoted by cold stress in rats or a chemogenetic activation in mice [15,18]. However, these experiments have limitations: Sánchez et al., 2009 [15] used the i.p. injection of a TRH-DE inhibitor that passes through the blood–brain barrier, making it impossible to pinpoint the anatomical site involved; Müller-Fielitz et al., 2017 [18] could not untangle the relevance for HPT axis activity of tanycytes end-feet from that of tanycyte TRH-DE or that of other tanycyte factors down-stream of Gαq/11 proteins.

Overexpression of TRH-DE activity in the ME reduced serum TSH concentration in response to cold exposure, likely because the availability of TRH to the pituitary was reduced. These data argue in favor of the idea that ME TRH-DE activity controls the phasic output of TRH into the portal vessels but do not rule out the impact of tanycyte end-feet’s physical interaction with portal capillaries. We next used the viral strategy to investigate the control of basal serum TSH concentration by ME TRH-DE.

### 4.4. Long-Term Alteration of ME TRH-DE Activity Has Either No or a Small Impact on Basal Serum Concentration of TSH and TH

In healthy adult mice housed in standard conditions, the KO of genes involved in the central regulation of the HPT axis generally upsets thyroid homeostasis while maintaining serum tT3 concentration inside circadian variation [67]. For example, global or PVH-specific deletion of *Trh* increases serum TSH concentration diminishes TSH biological activity, and leads to hypothyroidism [68,69]. Deletion in adult tanycytes of diacylglycerol lipase alpha—an enzyme that synthesizes an endocannabinoid that inhibits the release of TRH from nerve terminals in the ME [16]—increases the pituitary *Tshb* mRNA level and circulating free T4 concentration [70]. Global *Trhr* deletion does not affect serum TSH concentration but produces hypothyroidism [71,72]. Deletion of *Dio2* in the pituitary or in thyrotropes leads to high serum TSH concentration with low biological activity and normal serum TH concentrations [73,74]. *Thrb*^−/−^ mice are hyperthyroid with enhanced serum TSH concentration [75]. In contrast, the PVH-specific deletion of *Dio3* does not change the hypothalamic expression of *Trh* or serum concentrations of TSH, tT4, and tT3 in mice of both sexes at baseline [76]. Likewise, for *Trhde* KO mice, we expected a mild hyperthyroid status but HPT axis variables were not distinct from those of WT mice. Therefore, in standard conditions, the basal activity of the HPT axis seems resistant to long-term *Trhde* elimination, despite no evidence that TRH-DE hydrolytic activity can be substituted by that of another protein [21].

Nevertheless, in rats we obtained a more nuanced result. Except for a pair of experiments, we generally did not observe a significant effect of 3V AAV1-TRH-DE or AAV1-TRH-DEt injection over basal serum TSH concentration compared to AAV1-GFP injection. Neither *Trh* expression in the PVH, pituitary *Tshb* expression, total serum TH concentration, nor serum tT3/tT4 ratio—an indirect marker of TH biotransformation—nor body or WAT weights, nor expression of down-stream targets of TSH action were affected by 3V AAV1-TRH-DE or AAV1-TRH-DEt injection. The data were the same using different TSH immunoassays, limiting the possibility that TRH-induced TSH glycosylation [77], and thus immunodetection, might have masked changes in serum TSH concentration. We cannot however exclude the possibility that the biological activity of TSH was modified.

Another limitation of our approach in rats is that viral injections did not always change ME TRH-DE, possibly because variations in the injection site altered the efficiency of transduction, and/or because TRH concentration in ME extracellular space negatively regulates endogenous TRH-DE activity [18]. Thus, the conclusion that ME TRH-DE activity does not influence basal serum TSH concentration in rats could have been due to the way we analyzed the data, i.e., without separating successful injections from those that did not change ME TRH-DE activity, which might have reduced further statistical power. Indeed, when the output of multiple experiments was grouped, we found a small negative correlation between TRH-DE ME activity and serum TSH concentration, suggesting a small inhibitory influence of TRH-DE over basal serum TSH concentration in rats reared in standard conditions.

### 4.5. Long-Term Alteration of Trhde Expression Impacts Parameters That Might Control the Basal Activity of the HPT Axis in ME and Anterior Pituitary

Although we observed that serum TSH concentration was not or was minimally affected by manipulation of ME TRH-DE activity, we nevertheless detected two adaptations that might have contributed to the stability of serum TSH concentration, in addition to the short-loop positive feedback that TRH exerts on ME TRH-DE activity [18]. First, the number of vimentin-positive tanycyte ramifications in the ME was higher in *Trhde* KO mice than in WT mice; excess TRH concentration around β2-tanycyte end-feet might have activated a TRH-R1 transduction pathway that increases the coverage of portal vessels with tanycyte extensions [18], limiting TRH access to portal capillaries, and thus maintaining basal TSH concentration. However, a limitation of our work is that vimentin extensions do not necessarily reflect the real cytoplasmic tanycyte coverage of ventral vessels. Second, AAV1-TRH-DE 3V injection increased anterior pituitary *Trhr* expression, possibly because of a decrease in TRH concentration in the anterior pituitary extracellular space [52,53,78]. Increased expression of the receptor might have dampened the physiological consequence of a decrease in pituitary TRH availability. Although preliminary, these observations suggest that the impact of sustained modifications of ME TRH-DE activity on HPT axis activity is opposed by other events that change TRH secretion from nerve endings, and limit TRH entry in the portal capillaries or TRH action on thyrotropes.

### 4.6. Multiple Evidence Indicates That Trhde Expression Is Not Critical for the Control of the Basal Activity of the HPT Axis

Despite experimental evidence that ME TRH-DE is positioned in proximity to TRH buttons and portal vessels, and that ME TRH-DE activity controls the phasic output of TRH into the portal vessels, previous results had shown that an intravenous injection of a specific inhibitor of TRH-DE to adult male rats reared in standard conditions does not change basal serum TSH concentration [15]. Likewise, the i.p. injection of the TRH-DE inhibitor Glp-Asn-Pro-D-Tyr-D-TrpNH2 hardly changes basal TSH concentration in rats [79]. In addition, other data indicate that anterior pituitary TRH-DE activity is not critical for short-term control of TRH-induced TSH release from anterior pituitary cells [80]. Our data in mice and rats suggest that the long-term modification of ME TRH-DE activity has a limited impact on basal serum TSH and TH concentrations, probably because specific adaptations of TRH output and/or action limit the change in the basal concentration of serum TSH. In agreement, tanycyte Gαq/11 KO, which reduces ME TRH-DE activity, does not modify basal HPT axis activity [18]. Other data show that short-term ablation of the β-tanycyte layer (which is substituted by a GFAP-positive glial scar) in tamoxifen-inducible Rax-CreERxEno2-lsl-DTA mice promotes a small rise in serum TSH concentration while the long-term ablation does not [81]. Finally, genome-wide association studies indicate that the approx. 100 genes relevant for setting serum TSH levels in humans do not include *Trh*, *Trhr*, or *Trhde* [82].

## 5. Conclusions

When the activity of the hypophysiotropic TRH neurons is relatively stable, short-term or long-term changes in ME TRH-DE activity might not modify efficiently the amount of TRH reaching the anterior pituitary, or the efficiency of TRH at the pituitary level, because of either rapid or slow adaptations at the ME and/or pituitary level. However, when hypophysiotropic TRH neurons are burst-firing, these adaptations might be insufficient in front of the large amount of TRH released, making ME TRH-DE activity critical for regulating serum TSH concentration.

## Figures and Tables

**Figure 1 cells-14-00725-f001:**
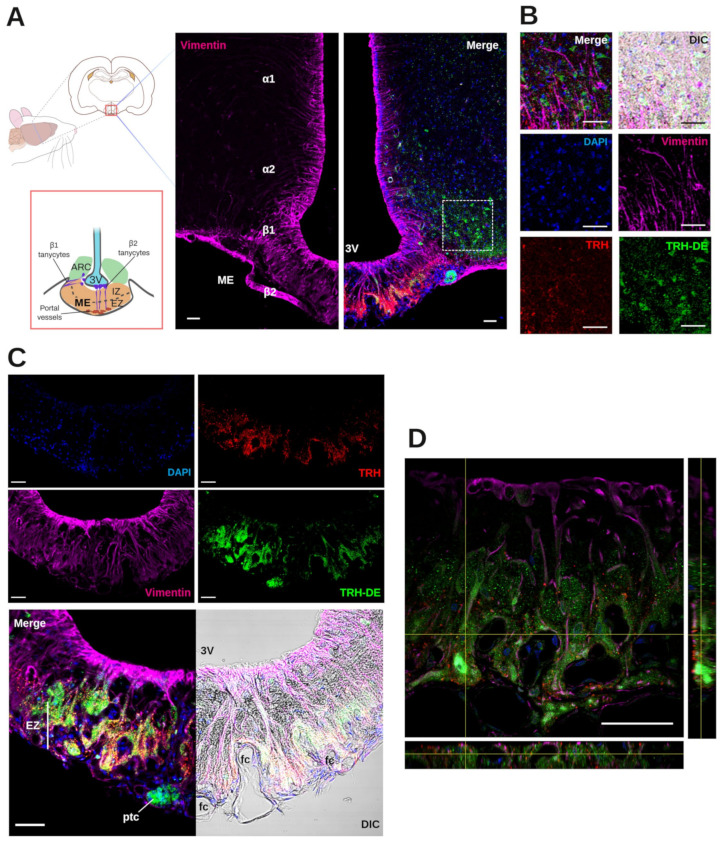
Distribution of TRH-DE immunoreactivity in basal medial hypothalamus of adult male rats: (**A**) Left panel: ME (lateral β1 and central β2) tanycytes form barriers between the cerebrospinal fluid and the ME parenchyma, or between the ME itself and the ARC parenchyma. Axons and terminal buttons from hypophysiotropic TRH neurons of the PVH are abundant in the external zone (EZ) of the ME (IZ: internal zone). Center and right panels: Photomicrograph of the basal medial hypothalamus showing the distribution of vimentin-expressing tanycytes (magenta, approximate position of subtypes in Greek letters) along the walls of the 3V, TRH-DE immunoreactivity (green), and their relative position to TRH terminals (red). The dotted square defines the localization of the higher-magnification images of the next panel. (**B**) TRH-DE is detected in cells of the ARC that are not tanycytes and in the ME. We also detected TRH-DE in pars tuberalis cells (ptc). (**C**) β2-tanycytes project a basal process into the EZ of the ME, from where fenestrated portal capillaries (fc) receive hypophysiotropic signals transported to the anterior pituitary. TRH-DE immunoreactivity can be traced along the intermedial/external zone of the ME, where it coincides with the apical part of the feet of vimentin-expressing β2-tanycytes, and with hypophysiotropic TRH terminal buttons, near fenestrated portal capillaries where neurons release TRH. (**D**) z-projection shows TRH-DE immunoreactivity proximal to TRH buttons in the external zone of the ME. Bar: 50 μm.

**Figure 2 cells-14-00725-f002:**
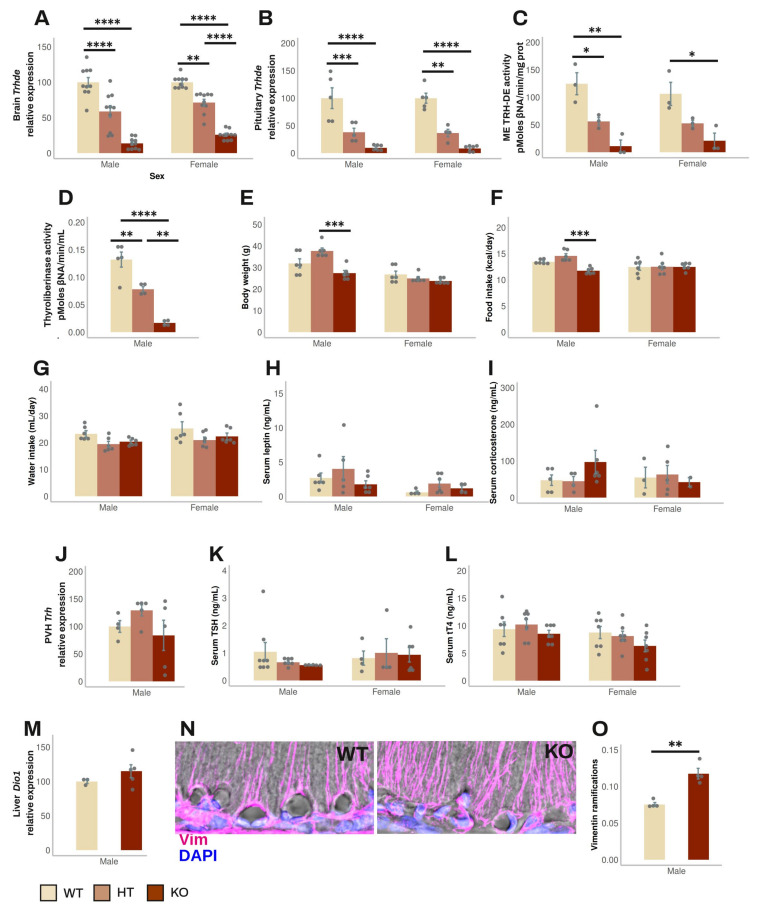
Effect of Trhde^tm1Lex^ alleles on the phenotype of young adult C57BL/6NJ mice maintained in standard conditions: (**A**) *Trhde* expression in brain. (**B**) *Trhde* expression in pituitary. (**C**) ME TRH-DE-specific activity. (**D**) Thyroliberinase-specific activity. (**E**) Body weight. (**F**) Food intake. (**G**) Water intake. (**H**) Serum leptin concentration. (**I**) Serum corticosterone concentration. (**J**) PVH *Trh* mRNA levels. (**K**) Serum TSH concentration. (**L**) Serum tT4 concentration. (**M**) *Dio1* mRNA levels in liver. (**N**) Photomicrographs of β2-tanycyte basal extensions. (**O**) Quantification of the number of vimentin (Vim) ramifications per mm^2^ of basal membrane in confocal slices. Data (mean ± SEM (n)) correspond to one cohort. Expression (mRNA levels) data are shown in percentage of WT mean taken as 100%; internal control mRNAs were *Actb* for panels A and M, *Ppia* for panel B, and *Hprt* for panel J. Statistical tests: two-way ANOVA with Tukey’s post hoc test in (**A**–**C**, **E**, and **F**); one-way ANOVA in (**D**); Mann–Whitney test in (**O**). *: *p* < 0.05; **: *p* < 0.01; ***: *p* < 0.001; ****: *p* < 0.0001.

**Figure 3 cells-14-00725-f003:**
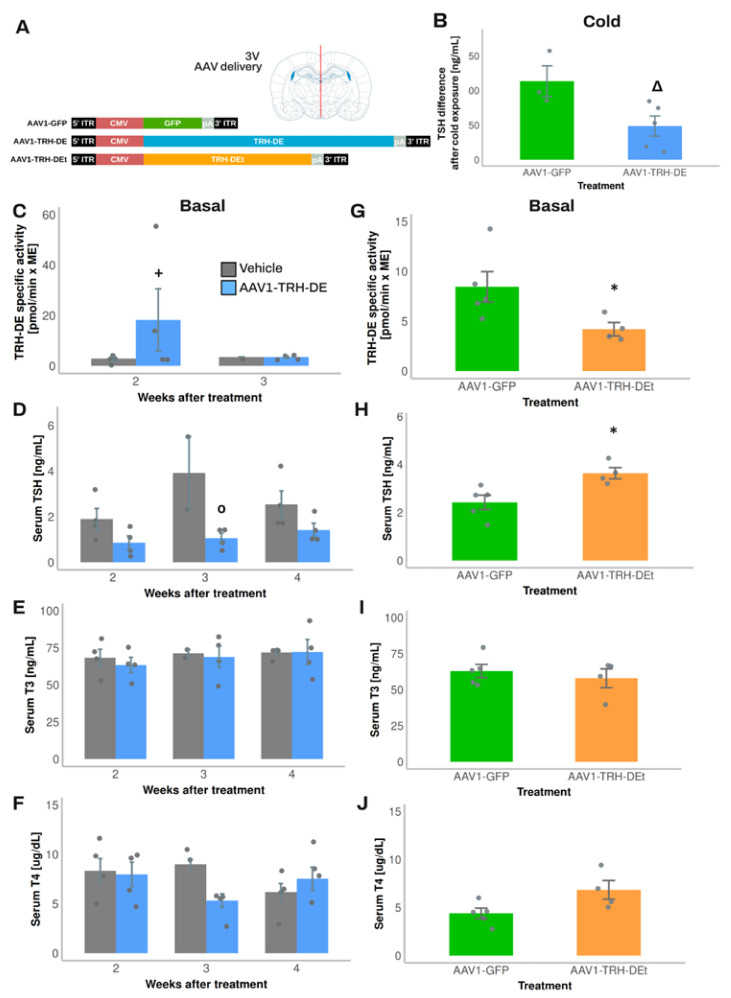
Effects of delivery of AAV1-TRH-DE or AAV1-TRH-DEt (8.75 × 10^10^ vg) into the 3V in serum TSH, tT4, and tT3 concentrations of young male Wistar rats maintained in standard conditions or cold-exposed: (**A**) Schematic representation of AAV1-GFP, AAV1 TRH-DE, and TRH-DEt expression vectors used in experiments and site of injection in adult rats. ITR: internal terminal repeats, CMV: human cytomegalovirus immediate early enhancer and promoter; pA: polyadenylation signal. Experiment 2—(**B**) one-hour cold-induced serum concentration of TSH measured 2 weeks after viral delivery of AAV1-GFP or AAV1-TRH-DE. Experiment 3—(**C**) temporal assessment of TRH-DE activity in ME, (**D**) serum TSH concentration (by RIA), (**E**) tT3 concentration, and (**F**) tT4 concentration 2–4 weeks after AAV1-TRH-DE (blue) or vehicle (gray) delivery. Experiment 4—effect of AAV1-TRH-DEt delivery (**G**), which decreased ME TRH-DE activity on (**H**) serum TSH, (**I**) tT3, or (**J**) tT4 concentrations. Data are mean ± SEM. Statistical tests: two-way ANOVA with Bonferroni’s post hoc test in (**B**,**D**); Student’s *t*-test in (**F**,**G**). *: *p* < 0.05 for vehicle vs. AAV1-TRH-DE at week 2; ^+^: *p* < 0.05 for AAV1-GFP vs. AAV1-TRH-DE; °: *p* < 0.01 for vehicle vs. AAV1-TRH-DE. Δ: *p* < 0.05 for AAV1-GFP vs. AAV1-TRH-DE.

**Figure 4 cells-14-00725-f004:**
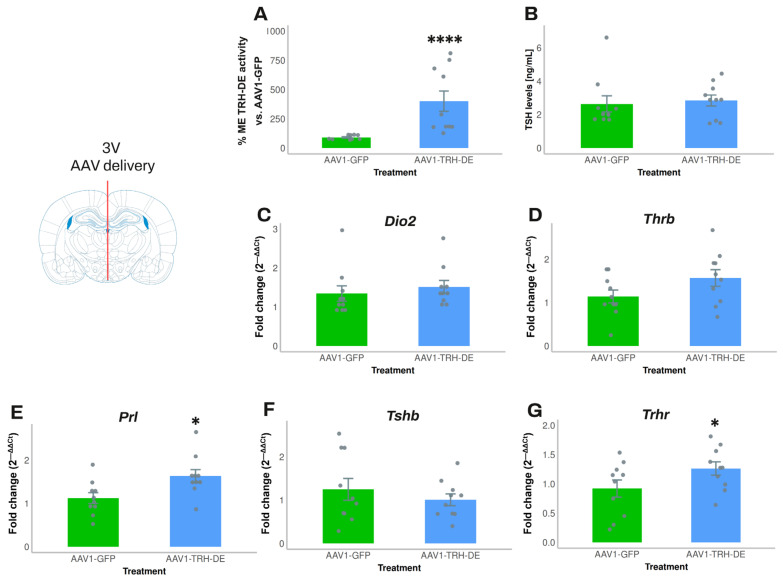
TRH-DE overexpression in the ME did not influence serum TSH concentration but modified the expression of some pituitary targets of TRH in young male Wistar rats maintained in standard conditions. Experiment 5—analysis 2 weeks after 3V injection of increased amount (8.75 × 10^10^ (n = 5) and 4.37 × 10^11^ vg (n = 5)) of AAV1-TRH-DE or of AAV1-GFP. (**A**) Percentage of ME TRH-DE activity relative to AAV1-GFP group. (**B**) Serum TSH concentration (by ELISA). (**C**) *Dio2*, (**D**) *Thrb*, (**E**) *Prl*, (**F**) *Tshb*, and (**G**) *Trhr* expression in the anterior pituitary. Data are mean ± SEM. Statistical tests: Mann–Whitney test in (**A**); Student’s *t*-test in (**E**,**G**). *: *p* < 0.05 and ****: *p* < 0.0001 for AAV1-GFP vs. AAV1-TRH-DE.

**Figure 5 cells-14-00725-f005:**
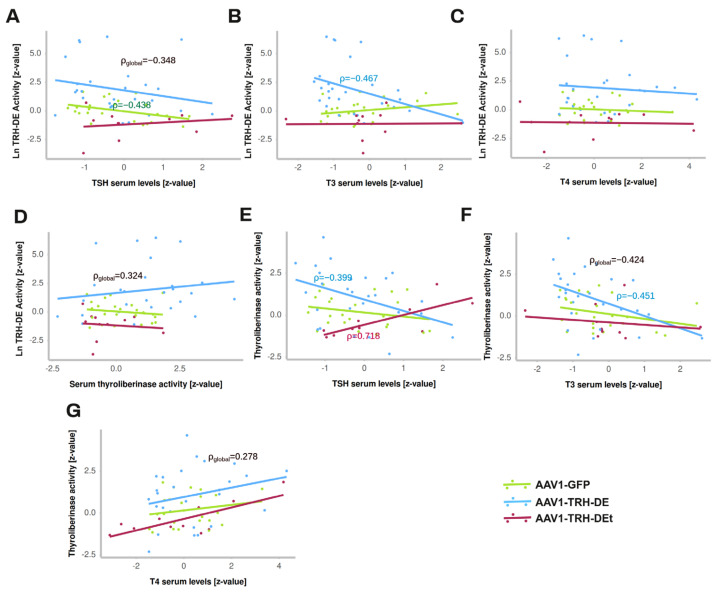
Correlations between hormonal values in HPT axis and TRH-DE activity of young male Wistar rats maintained in standard conditions. The outputs from experiments 3, 4, 5, and 6 were grouped and analyzed to find correlations between variables measured 2 weeks after viral delivery of AAV1-GFP (4.37 × 10^11^ vg (n = 24)), or AAV1-TRH-DE (8.75 × 10^10^ (n = 22) and 4.37 × 10^11^ vg (n = 5)), or AAV1-TRH-DEt (4.37 × 10^11^ vg (n = 11)): (**A**) Correlations between ME TRH-DE activity and serum concentrations of TSH (by ELISA, RIA, or Milliplex), (**B**) tT3, (**C**) tT4, and (**D**) thyroliberinase activity. (**E**) Correlations between thyroliberinase activity and serum concentrations of TSH, (**F**) tT3, and (**G**) tT4.

**Figure 6 cells-14-00725-f006:**
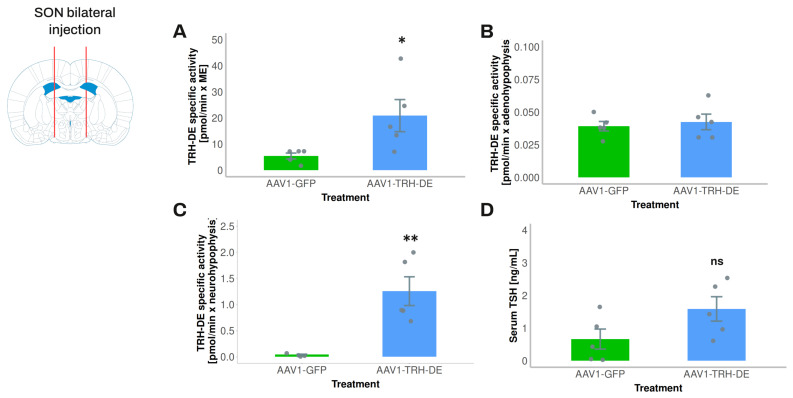
TRH-DE overexpression in the posterior pituitary did not affect serum TSH concentration of young male Wistar rats maintained in standard conditions. Experiment 7—(**A**) two weeks after bilateral injection of AAV1-TRH-DE (4.37 × 10^9^ vg) in the SON, we detected enhanced TRH-DE activity in the ME and (**C**) the posterior pituitary, (**B**) while not in the anterior pituitary. (**D**) AAV1-TRH-DE injection did not change serum TSH concentration (by ELISA) when compared to AAV1-GFP-injected controls. Data are mean ± SEM. Statistical tests: Mann–Whitney test in (**A**,**C**); Student’s *t*-test in (**D**). *: *p* < 0.05 or **: *p* < 0.01 for AAV1-GFP vs. AAV1-TRH-DE; ns: not statistically significant.

## Data Availability

The original contributions presented in this study are included in the article/Appendix A. Further inquiries can be directed to the corresponding author.

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
