# Peer review of "Hypothalamic Median Eminence Thyrotropin-Releasing Hormone-Degrading Ectoenzyme Activity Is Dispensable for Basal Thyroid Axis Activity in Lean Rodents"

_cells, 2025, doi:10.3390/cells14100725_

Round 1
Reviewer 1 Report
Comments and Suggestions for Authors
The manuscript presents original data concerning the release of TRH into the bloodstream and its relationship with the median eminence. The illustrations are excellent and represent the data available well. I would add a few comments and suggestions to add a more presentable to the readers, and probably would add beauty to an already good manuscript, as follows:
- In the Introduction Chapter between lines 38-40, it would be good to add an illustration to make it clear to the readers what has been said in those lines. A colorful illustration would didactically represent the written phrase.
- Between lines 51-54 work the same, an illustration representing the layers of the median eminence would make clear to the reader the composition of such structure.
- What is the rationale for searching for the differences between rats and mice? Please, make a clearer statement for that.
- In the Materials and Methods Chapter it would be more easier to understand what the authors are trying to tell, if they have been more specific in the description of the sexes used in the experiments. Mice "the same sex". Line 103. Which sex? The authors culled the pupps to 10 animals, and used 5 males. Why the necessity to cull the pupps? Why the authors did not do the experiments in females as well? The lines 107-111, for me are not clear at all. Please, be more specific and detail the description of the sexes used and why, for both mice and rats.
Reviewer 2 Report
Comments and Suggestions for Authors
In this manuscript, Rodríguez-Rodríguez et al. investigates the contribution of the TRH-degrading ectoenzyme (TRH-DE) to the basal control of the hypothalamus-pituitary-thyroid axis.
Overall, I find the central question very interesting and very relevant in the fine tuning of HPT axis activity. While the majority of the presented data shows the lack of effect of TRH-DE manipulation, I find it very thorough and meticulous. This manuscript lays a strong foundation for future research on the role of TRH-DE.
One minor change I would recommend that could help the accessibility of the manuscript. In the results section, the references to experiments 1-7 are very technical and hard to follow. Some simple nomenclature instead of the numbers would help. Also, it appears that experiment 3 is presented after 6 and experiment 7 is mentioned first in the results.
Reviewer 3 Report
Comments and Suggestions for Authors
In the manuscript “Hypothalamic Median Eminence Thyrotropin-Releasing Hormone-Degrading Ectoenzyme Activity is Dispensable for Basal Thyroid Axis Activity in Lean Rodents” by Rodríguez-Rodríguez et al. the authors presented a huge investigation of the relevance of TRH-DE for the basal activity of the HPT axis. The authors investigate global TRH-DE knockout mice as well as the tanycyte-specific overexpression of an active TRH-DE and a truncated dominant negative form of TRH-DE (TRH-DEt) in rats.
To transduce the tanycytic expression of TRH-DE and TRH-DEt, the authors tested different AAV serotypes to optimize the tanycyte-specific transduction. This is a very important information for using this technique in future experiments targeting tanycytes in rats.
The authors convincingly demonstrated the modulation of the TRH-DE activity. However, the physiological outcomes under basal conditions were minor and transient, which indicates a high adaption of the HPT axis. Unfortunately, the authors cannot identify the compensatory mechanisms when TRH-DE activity is regulated.
The manuscript gives new and important insight into the functions of TRH-DE especially in tanycytes. However, there are major and some minor points to address, which have to be revised by the authors.
- The authors claim that only in Experiment 1 the basal TSH was different between AAV1-GFP and AAV1-TRH-DE over time (Fig.3C). There are some inconsistencies in this illustration. For example, 2 data points are missing in the AAV-GFP group at 3 weeks. In addition, one data point is missing in the AAV1-TRH-DE group at 2 weeks. Why the datapoints were excluded?
- What is much more important, however, is that in the AAV1-TRH-DE groups after 2 and 3 weeks the bar (mean) and the error bar cannot match the plotted data points. How does this come about? A critical review of the statistics and the statements made is therefore unavoidable. In my opinion, this point is absolutely critical, as this is a result discussed in the manuscript.
- The statistics must be checked in Figure 3c. A 2-way ANOVA must be performed here. For all analyses, the statistical parameters as well as the results T-, F-value should be given as well as the results of the posttests.
- Figure 3 shows two different significances. The asterisk is given as the significance between the vehicle and AAV1-TRH-DE. However, this is also used for the comparison between vehicle and AAV1-TRH-DEt (3F and 3G). This must be clearly shown in the legend. In addition, it looks as if the circle used (3C) represents the statistical result in a post-test. However, this appears to represent the initial result of the 2-way ANOVA (according to the legend). This needs to be checked and the legend or figure needs to be adjusted accordingly.
- What is showing mean, median, SE or SEM. Please clarify it in the legend.
- The authors speculate about a change in TSH activity. It is possible to validate TSH activity in plasma using a cAMP assay performed in TSHR-expressing cells. This can be used to test this hypothesis.
- In cold stress Rats are transduced with TRH-DE in tanycytes does not increase TSH in serum. How do the completed TRH-DE knockout mice respond to cold stress? Is there a change in TSH levels compared to WT littermates to confirm a change under stimulation of the HPT axis?
- For better clarity in the manuscript, the figures and sub-figures should follow the text. At the moment there is a lot of jumping between and within the characters. The reaction to cold is discussed at the beginning of the results, which refers to Figure 5. This should be adjusted.
- The authors compared 7 different experiments in rats. It would be good to explain the differences in detail in the main method part. By reading the manuscript the reader gets lost about the differences in the protocols.
- In the Methods the authors describe the measurements of hormones like prolactin and Il-6 these measurements are missing in the results. Otherwise, the part has to be deleted.
- The scale bars are missing in 1B, 1D, and sup. Fig. 3. Sometimes the color code is missing. Please check all microscopic pictures.
- Line 221: Is the exposure time of 4 seconds correct? That is quite long. The PFA fixation reduces the GFP fluorescence. Since the GFP signal appears to be weak, the GFP must be stained with a GFP antibody to make a real statement about the transduction efficiency and specificity. This should be done during the evaluation of the AAV transductions.
Minor:
- Line 391: Please show or mark the pars tuberalis cells in Figure 1C.
- Line 537: ° is underlined. Please check the whole manuscript.
- Line 66: Please include a reference for the expression of the genes in tanycytes e.g. Campbell et al. 2017, Chandrasekar et al. 2023, or your review from 2019.
- Line 104 und 111: It would be good to specify the light phase here. It would be better to specify it in hours after the light change.
- Line 235: what volume was injected?
- Line 265: 12000xg
- In Sup Fig 2c: the left ARC is in the wrong position
- Line 774: THR-beta knockouts are not hypothyroid. They are hyperthyroid since the disturbed feedback. Please correct the statement.
- Line 62: The authors mention a possible retrograde transport of T3 by hypophysiotropic TRH neurons as a conclusive fact. However, in the cited paper by Salas-Lucia et al 2024 there are only hints but no conclusive experiments for this hypothesis. As discussed by Salas-Lucia et al, a weakened formulation would be important in my view.
- Reference 23: Please refer also to the database https://www.mmrrc.org to avoid confusion, as the TRH-DE KO mouse is not directly mentioned in the publication.
Round 2
Reviewer 3 Report
Comments and Suggestions for Authors
The points previously raised have been addressed, and I am satisfied with the responses of the authors.
Unfortunately, the new Sup. Table 6 contains incorrect labels. The cold experiment, for example, is now Fig. 3B, not 5H. Therefore, the table also requires correction for Experiments 3 and 4, along with careful checking of all parameters.
The authors now indicate that data points were not excluded (line 366), and only whole animals with health problems were removed from the experiment. Together with the additional statistics table (Sup. Tab. 6), I now identify additional issues with missing data points and statistics.
In the new Fig. 3H (TSH serum) and Fig. 3J (Serum T4), the group AAV1-TRH-DEt includes 4 data points, while Fig. 3I shows only 3 data points, obviously from the same plasma samples. If no data point was excluded, as mentioned in line 366, why is one point missing? In the statistics of new Fig. 3G (in Sup. Tab. 3F), the df is 9, indicating a total sample size of 11, but there are only 10 animals in the experiment; please check the statistics. Why does the statistical analysis show one additional sample?
In the new Fig. 3H (compared to the old Fig. and Sup. Tab. 6), the indication of a significant difference (star) is now missing. Please include it in the figure.
In the new Fig. 4B, there is an issue with the mean and the data points in the AAV1-GFP group; the mean does not align with the data points, as only 1 point is higher than the mean. Please verify the figure and data points.
In Fig. 4, the statistics for E and G are unclear. For Prl (Fig. 4E), the df is reported as 17 (Sup. Tab. 6), suggesting a total sample size of 19; however, only 18 data points are displayed. How many animals are included? Sup. Table 2 indicates a group of 6 animals per group, while the legend specifies an n of 5 per group, but only 18 points are shown. In line 366, the authors state that no points were removed. How is it possible that in Fig. 4G, all 20 data points (as indicated in the legend) are shown when only animals were excluded? For Trhr, the PCR was run for 20 samples. Please clarify this discrepancy.
For Trhr (Fig. 4G), the df is 17 (Sup. Tab. 6), indicating a total sample size of 19; however, 20 data points are shown. Why was the statistic performed with only 19 samples?
Please carefully check the statistics for all figures again to prevent errors before publishing.
Round 3
Reviewer 3 Report
Comments and Suggestions for Authors
All points are corrected.
Many thanks for the imporant results.